# A Redesigned Method for CNP-Synchronized In Vitro Maturation Inhibits Oxidative Stress and Apoptosis in Cumulus-Oocyte Complexes and Improves the Developmental Potential of Porcine Oocytes

**DOI:** 10.3390/genes14101885

**Published:** 2023-09-28

**Authors:** Jinlun Lu, Min Guo, Xiaodong Wang, Rui Wang, Guangyin Xi, Lei An, Jianhui Tian, Meiqiang Chu

**Affiliations:** 1State Key Laboratory of Animal Biotech Breeding, National Engineering Laboratory for Animal Breeding, Key Laboratory of Animal Genetics, Breeding and Reproduction of the Ministry of Agriculture and Rural Affairs, College of Animal Science and Technology, China Agricultural University, No. 2 Yuanmingyuan West Road, Beijing 100193, China; 2College of Agriculture and Forestry Science, Linyi University, Linyi 276000, China

**Keywords:** C-type natriuretic peptide, meiotic arrest, in vitro maturation, developmental potential, porcine oocytes

## Abstract

In vitro embryo production depends on high-quality oocytes. Compared with in vivo matured oocytes, in vitro oocytes undergo precocious meiotic resumption, thus compromising oocyte quality. C-type natriuretic peptide (CNP) is a follicular factor maintaining meiotic arrest. Thus, CNP-pretreatment has been widely used to improve the in vitro maturation (IVM) of oocytes in many species. However, the efficacy of this strategy has remained unsatisfactory in porcine oocytes. Here, by determining the functional concentration and dynamics of CNP in inhibiting spontaneous meiotic resumption, we improved the current IVM system of porcine oocytes. Our results indicate that although the beneficial effect of the CNP pre-IVM strategy is common among species, the detailed method may be largely divergent among them and needs to be redesigned specifically for each one. Focusing on the overlooked role of cumulus cells surrounding the oocytes, we also explore the mechanisms relevant to their beneficial effect. In addition to oocytes per se, the enhanced anti-apoptotic and anti-oxidative gene expression in cumulus cells may contribute considerably to improved oocyte quality. These findings not only emphasize the importance of screening the technical parameters of the CNP pre-IVM strategy for specific species, but also highlight the critical supporting role of cumulus cells in this promising strategy.

## 1. Introduction

The acquisition of development potential is largely dependent on well-orchestrated oocyte maturation, which involves two relatively independent but related processes, i.e., nuclear and cytoplasmic maturation [1]. The coordination and synchronization of nuclear and cytoplasmic maturation is the prerequisite for acquiring the developmental potential of in vivo oocytes [2,3]. In contrast, the in vitro isolation of oocytes from the natural follicular environment leads to a premature decline in intra-oocyte cyclic adenosine monophosphate (cAMP) concentrations and thus triggers spontaneous and precocious meiotic resumption [4,5], which in turn leads to a nuclear-cytoplasmic asynchrony and ultimately compromises the developmental potential of the oocytes. Thus, spontaneous oocyte meiotic resumption has been thought to be the main obstacles to the success of in vitro embryo production [6]. One widely applied strategy to overcome this asynchrony is to utilize conditions that maintain meiotic arrest and delay spontaneous resumption before in vitro maturation (IVM) [7,8,9,10]. A previous study by our research group, using bovine oocytes as the model, has established a natural factor synchronized in vitro oocyte maturation (NFSOM) system, which recapitulates the in vivo follicular meiotic arrest condition via C-type natriuretic peptide (CNP) pretreatment, thus improving the developmental potential of mature bovine oocytes and resulting in higher efficiency of in vitro embryo production [11].

Over the past decades, CNP has emerged as a critical regulator of the progression of nuclear maturation by maintaining intra-oocyte cAMP levels in many species [12,13,14,15]. In growing follicles, CNP secreted from the mural granulosa cells acts on its guanylyl cyclase natriuretic peptide receptor 2 (NPR2), located on the cumulus cells (CCs), and then stimulates the generation of cyclic guanosine monophosphate (cGMP) [12], which is then transferred to the oocyte through gap junctions. Increased intra-oocyte cGMP inhibits the activity of phosphodiesterase 3A (PDE3A), thus resulting in the cAMP hydrolysis inhibition and the maintenance of meiotic arrest until luteinizing hormone-induced oocyte meiotic resumption occurs [12,16].

The CNP-synchronized IVM strategy has been well-accepted and applied to update the conventional IVM method of mouse, bovine, sheep, and goat oocytes [17,18,19,20]. Although the CNP-pretreatment strategy has been used to modify the IVM method of porcine oocytes, the beneficial effect was limited to oocytes from small (3–4 mm) and medium (4–6 mm) follicles. Of note, the developmental potential of oocytes from large (6–8 mm) follicles, accounting for a large proportion of the collected oocytes, was unexpectedly decreased due to CNP-pretreatment [21]. This finding seems to challenge the concept of the CNP-synchronized IVM strategy. However, the adverse effect specific to fully grown porcine oocytes led us to question whether or not the detailed method was reasonable and in need of redesign.

Oocyte maturation is a complex process regulated by many internal and external conditions [22]. Among these, the intracellular redox homeostasis and apoptotic regulation in cumulus cells are very important factors for oocyte maturation [23,24]. Dysregulated redox homeostasis and apoptosis were thought to be tightly coupled with lower oocyte quality after IVM [23,25,26]. Thus, the status of cumulus cells is an important indicator of oocyte quality. However, the effect of CNP on the cumulus cell status has been largely overlooked. In addition, a series of heterochromatin-related epigenetic landmarks in oocytes are established upon meiosis, which is critical to safeguard oocyte quality and subsequent embryogenesis [27,28]. Defects in heterochromatin-related epigenetic modifications may contribute to compromised oocyte quality [29,30].

In this study, we tested the efficiency and feasibility of the CNP pre-IVM strategy to improve the porcine oocyte IVM system and explored the possible mechanisms of the beneficial effects. We also focused on the effect of CNP on the status of cumulus cells and heterochromatin-related epigenetic modifications, which are critical to oocyte quality, but which have often been overlooked in previous studies.

## 2. Materials and Methods

Unless otherwise stated, all chemicals and reagents were purchased from Sigma-Aldrich (St. Louis, MO, USA). 

### 2.1. Isolation and Collection of Cumulus-Oocyte-Complexes (COCs), Denuded Oocytes (DOs), Cumulus Cells (CCs) and Mural Granulosa Cells (MGCs)

Porcine ovaries were acquired from commercially slaughtered prepubertal gilts at a local abattoir. They were promptly transported to the laboratory within 3 h, maintaining a temperature range of 28–30 °C while immersed in a saline solution (0.9% NaCl) containing antibiotics. Healthy follicles of 3–8 mm in diameter were aspirated to recover COCs using an 18-gauge needle attached to a 20 mL syringe. Following a brief incubation at 38.5 °C, the COCs within the follicular fluid were subjected to three washes using HEPES-buffered Tyrodes solution (TL-HEPES) supplemented with 0.3% (*w*/*v*) bovine serum albumin (BSA). Only COCs displaying uniform ooplasm and multilayered cumulus were selected for IVM. DOs were obtained by gentle pipetting into a PBS medium containing 0.5% (*w*/*v*) hyaluronidase. The collected CCs and MGCs were promptly frozen in liquid nitrogen and stored at −80 °C for subsequent analysis. 

### 2.2. In Vitro Maturation of Porcine COCs

To assess the impact of CNP on the nuclear meiosis of porcine oocytes, the dose-response and time-dependent effect of CNP pretreatment on meiotic maturation were analyzed. The pretreatment IVM medium consisted of TCM-199 (Gibco, Grand Island, NY, USA), 10% (*v*/*v*) porcine follicular fluid, 0.6 mM cysteine, 75 mg/L of penicillin, and 50 mg/L of streptomycin. The IVM medium was supplemented with 10 IU/mL of hCG (Chorulon, Intervet Australia Pty Limited, Victoria, Australia), 10 IU/mL of PMSG (Folligon, Intervet Australia Pty Limited), and 10 ng/mL of epidermal growth factor (EGF), based on the pretreatment medium. Selected COCs were subjected to three washes with the specified medium and subsequently transferred to droplets, each containing 60 μL of pre-equilibrated pre-IVM or IVM medium, all situated under 3 mL of mineral oil in a 35 mm Petri culture dish containing 15 to 20 COCs per droplet. The COCs within the droplets were cultured at 38.5 °C in a humidified environment with 5% CO_2_. According to the pre-IVM method, COCs in the CNP pre-IVM group underwent culture in IVM droplets for either 17 or 41 h following the initial 24 h pre-IVM phase. In contrast, COCs in the conventional IVM group were cultured in IVM droplets for the full 41 h duration. 

### 2.3. Assessment of Nuclear Maturation

At different timepoints after pre-IVM culture, the oocytes were mechanically denuded. The DOs were then fixed in 4% paraformaldehyde (PFA) for 1 h and permeabilized in 0.5% PBST (0.5% Triton-100 (*v*/*v*) in PBS containing 0.1% (*w*/*v*) polyvinyl alcohol (PVA)). They were then stained with DAPI for 5 min. The stained DOs were placed on a glass slide, which was then covered with a coverslip. Images of the stained DOs were captured and evaluated. The percentages of the oocytes at the germinal vesicle (GV) and meiosis II (MII) stages in each experimental group were calculated. 

### 2.4. Parthenogenetic Activation (PA)

PA served as an alternative to in vitro fertilization to assess the developmental potential of porcine oocytes because of the possible effects of boar semen. After CCs removal, mature MII oocytes underwent three washes in an activation medium (comprising 0.28 mol/L mannitol; 0.01% polyvinyl alcohol; 0.05 mmol/L HEPES; 0.1 mmol/L CaCl_2_·2H_2_O; and 0.1 mmol/L MgCl_2_) before activation. PA was achieved via a 1.5 kV/cm direct pulse for 30 microseconds using a BTX Electro-Cell Manipulator 2001 (BTX, San Diego, CA, USA). Post-activation, the oocytes underwent three washes in PZM-3 medium containing 10 μg/mL cycloheximide (CHX) and 5 μg/mL cytochalasin (CB). Subsequently, the oocytes were cultured in PZM-3 supplemented with CB and CHX for 4 h, followed by three washes and culturing in PZM-3 under mineral oil in a 5% CO_2_ environment at 38.5 °C for 7 days. The cleavage and blastocyst rate were measured on days 2 and 7, respectively.

### 2.5. Immunofluorescence Staining (IF)

The DOs were initially fixed in 4% PFA for 1 h, permeabilized with 0.5% PBST, and subsequently blocked with 1%(*w*/*v*) BSA in 0.5% PBST for 6 h. For NPR2, H3K9me3, or H3K27me3 staining, the oocytes were incubated overnight at 4 °C with specific antibodies (NPR2: 1:50; Santa Cruz Biotechnology, Santa Cruz, CA, USA; H3K9me3 or H3K27me3: 1:1000; Millipore, Bedford, MA, USA). In the case of 5-methyl-cytosine (5mC) or 5-hydroxymethyl-cytosine (5hmC) staining, the permeabilized oocytes underwent denaturation in 4 N HCl for 20 min, neutralization with 100 mM Tris-HCl (pH 8.5) for 15 min, followed by blocking with 1% (*w*/*v*) BSA in 0.5% PBST overnight at 4 °C, and incubated with primary antibody (5mC and 5hmC: 1:250; Active Motif, Carlsbad, CA, USA) for 1 h at room temperature. After three washes, the DOs were incubated with secondary antibodies (488 goat anti-mouse IgG or 594 goat anti-rabbit IgG: 1:1000; Thermo Fisher Scientific Company, Waltham, MA, USA) for 1 h, with the nuclei counterstained using DAPI. The fluorescence was visualized and quantified using an inverted epifluorescence microscope (IX71; Olympus, Japan), with relative fluorescence intensity analyzed via ImageJ software (Version: 1.53a, National Institutes of Health, Bethesda, MD, USA). The experiments were conducted in triplicate.

### 2.6. Detection of Intracellular Reactive Oxygen Species (ROS) in the Oocytes

The intracellular ROS content within the oocytes was assessed utilizing a ROS detection assay kit (Beyotime Biotechnology, Shanghai, China), following the manufacturer’s instructions. Briefly, the oocytes were incubated in TCM199 medium containing 10 μM 2′,7′-dichlorofluorescin diacetate (DCFH-DA) for 30 min at room temperature. Subsequent to washing with 0.1% PBS-PVA, the oocytes were examined under a fluorescence microscope, with the same scanning settings used for all the groups. The fluorescence intensities of the oocytes were quantified using Image J software. The experiments were repeated at least three times.

### 2.7. Detection of Early Apoptosis by Annexin-V Staining in the Oocytes

The early apoptosis of the oocytes was detected using an Annexin-V staining kit (Beyotime Biotechnology, Shanghai, China). The viable DOs were washed and stained with 100 μL of a binding buffer containing 5 μL of Annexin-V-FITC for 20 min in the dark, according to the manufacturer’s instructions. After washing with PBS-PVA, the fluorescence signals were captured. The experiments were repeated at three times; 10–15 oocytes were included in each detection.

### 2.8. Terminal Deoxynucleotidyl Transferase-Mediated dUTP Nick-End Labeling (TUNEL) Assay

The apoptotic status of the COCs was evaluated using a TUNEL assay kit (Beyotime Bio-technology, Shanghai, China). The COCs were fixed with 4% PFA, permeabilized in 0.5% PBST for 20 min, and incubated in 0.5% BSA- PBS-PVA at room temperature. The COCs were then incubated in the TUNEL mixture for 1 h at 37 °C in the dark. After staining, the COCs were counterstained with DAPI. The labeled COCs were mounted on glass slides and imaged. The apoptosis rate was calculated as follows: apoptosis rate = (the number of TUNEL-positive cells/total cell number) × 100%. Image J software was employed for analysis. The experiments were conducted in triplicate.

### 2.9. RNA Extraction and Real-Time Reverse Transcription Quantitative PCR (RT-qPCR) Analysis

Following the manufacturer’s recommendations, TRIzol (Invitrogen) was used to extracted the total RNA from the pooled samples from the MGCs, CCs, and DOs, respectively. DNase (Vazyme Biotech, Nanjing, China) was applied to remove the residual genomic DNA before reverse transcription (RT). Approximately 0.5 μg of extracted RNA was used for each RT reaction. The complementary DNA was stored at −20 °C until use. RT-qPCR reactions were performed using a Bio-Rad CFX96 Real-Time PCR System with SosoFast EvaGreen Supermix (Bio-Rad Laboratories). The amplification protocol included an initial denaturation process at 95 °C for 30 s, followed by 40 cycles consisting of 5 s of denaturation at 95 °C and 5 s of annealing/extension at 60 °C. The negative controls were reactions without RT or the substitution of RNA samples with DEPC water (to detect any DNA contamination). The relative fold change of the mRNA transcripts was analyzed using the 2^−ΔΔCt^ method, with *RPL19* or *GAPDH* mRNA serving as reference genes. The primers are listed in Table 1. Each experiment was performed in triplicate.

### 2.10. Western Blot Analysis

The pooled samples from the MGCs, CCs, and DOs were lysed in Laemmli sample buffer (consisting of SDS sample buffer with β-mercaptoethanol) and subsequently heated to 100 °C for 5 min. A total of 10 μg cellular protein from each sample was separated by SDS-PAGE and electrophoretically transferred onto the PVDF membranes. Following transfer, the membranes were blocked using TBST (TBS containing 0.1% Tween 20) containing 5% non-fat milk for 1 h. Subsequently, the membranes were incubated overnight at 4 °C with a rabbit anti-NPR2 antibody (1:300, Santa Cruz Biotechnology, Santa Cruz, CA, USA) and a mouse monoclonal anti-β-GAPDH antibody (1:5000). After three washes with TBST, the membranes were further incubated with HRP-conjugated Goat anti-Rabbit and Goat anti-Mouse IgG (1:8000; ZSGB-Bio, Beijing, China) for 1 h at room temperature. The chemiluminescent signal was captured using a Tanon-5200 chemiluminescence imaging system (Tanon, Shanghai, China). All experiments were conducted independently, no fewer than three times.

### 2.11. Statistical Analysis

Statistical analyses were performed using SPSS 22.0 (SPSS, Chicago, IL, USA). Differences between groups were assessed using Turkey’s test for multiple means comparison. All data are presented as mean ± SEM. A *p*-value < 0.05 was considered statistically significant.

## 3. Results

### 3.1. The Expression Pattern of CNP and NPR2 in Porcine Ovarian Follicles

We first determined the expression patterns of CNP and its exclusive receptor NPR2 in the porcine ovarian follicles. The encoding mRNA of CNP, i.e., natriuretic peptide precursor C (*NPPC*) and *NPR2* were detected in the MGCs, CCs, and oocytes isolated from the preovulatory follicles. The *NPR2* mRNA was preferentially expressed in the CCs, whereas the *NPPC* mRNA level was significantly higher in the MGCs than in the CCs and the oocytes (Figure 1A). Of note, although the previous study by our research group reported that the oocyte membrane-localized NPR2 also contributes to the meiotic arrest of the bovine oocytes [11], the result from the present study indicated that notable NPR2 expression can be detected in the MGCs and CCs (Figure 1B,C), but not in the porcine oocyte membrane (Figure 1B,D), although *NPR2* mRNA was detected at low levels in the oocytes (Figure 1A), which highlighted the importance of NPR2 expression in the CCs for maintaining the meiotic arrest of the porcine oocytes. 

### 3.2. CNP Exposure of In Vitro Cultured COCs Maintains the Meiotic Arrest of Porcine Oocytes

It has been well-established that in vitro cultured oocytes undergo spontaneous and precocious meiotic resumption, thus impairing oocyte maturation. We therefore wanted to determine whether the exogenous supplementation of CNP could recapitulate in an in vivo follicular environment that maintains oocyte meiotic arrest. To test this theory, we exposed in vitro cultured COCs to CNP and measured the percentage of germinal vesicle breakdown (GVBD) that occurred as oocyte meiotic resumption (Figure 2A). Our results showed that CNP inhibited meiotic resumption in a dose-dependent manner, reaching its maximal effect at 200 nM (69.4% vs. 41.9%) (Figure 2B). Furthermore, dynamic analyses showed that 200 nM CNP exhibited a time-dependent inhibitory effect on preserving meiotic arrest up to approximately 24 h (Figure 2C). In addition, we found that vasopressin [31], an effective NPR2 inhibitor, abolished the inhibitory impact of CNP on meiotic resumption in porcine oocytes (Figure 2D), further emphasizing the critical role of NPR2 in mediating CNP-induced meiotic arrest in porcine oocytes.

### 3.3. CNP-Based Biphasic IVM System Improves the Developmental Potential of In Vitro Matured Porcine Oocytes

Our group has proposed a CNP-based pre-IVM strategy to improve bovine oocyte developmental potential via enhancing the synchronicity of nuclear and cytoplasmic maturation [11]. Although this method has been well-accepted among different species, the efficacy seems to be unsatisfactory in porcine oocytes. CNP pre-IVM, followed by the conventional IVM, led to a significant decrease in the developmental potential of the oocytes from large follicles (6–8 mm) [21]. This fact, together with the relatively longer duration of conventional porcine IVM per se, led us to postulate that porcine oocytes, especially fully grown oocytes, may be more sensitive to extended IVM duration than those of other species. Therefore, we developed two CNP-based biphasic IVM systems, with or without an extended interval, based on the selected concentration and interval, i.e., the COCs were pretreated with 200 nM CNP for 24 h, followed by 41 h or 17 h (equal to the conventional IVM duration in total) IVM (Figure 3A). Using parthenogenetic embryos, whose developmental performance is largely dependent on oocyte quality, as the model, we tested the preimplantation development following different biphasic IVM systems. Our results showed that oocytes from the CNP-based biphasic IVM system exhibited a higher, although not significant, cleavage rate, as well as a significantly higher blastocyst rate after PA, than those undergoing conventional IVM. In contrast, the extended IVM system, in agreement with previous published CNP pre-IVM system results [21], exhibited no beneficial effects on the oocyte developmental potential. These results indicated that the CNP-based biphasic IVM system was also beneficial to porcine oocytes, similar to the results obtained for previously tested species [11,17,18,19,20].

### 3.4. CNP Alleviates Oxidative Stress and Early Apoptosis in Porcine Oocytes

Next, we attempted to determine the mechanism relevant to the beneficial effect of the CNP-based biphasic IVM system. Previous studies have showed that standard in vitro culture conditions lead to oxidative stress and an imbalanced antioxidant defense system [32], and protecting oocytes against oxidative stress during IVM can improve their developmental potential. Therefore, we hypothesized that a CNP-based biphasic IVM system could enhance the resistance of porcine oocytes to oxidative stress. To test this theory, we assessed intracellular ROS levels using DCFH-DA. Quantitative analysis revealed that oocytes subjected to CNP-based biphasic IVM exhibited significantly lower relative intracellular ROS levels compared to those undergoing the 41 h conventional IVM treatment. Of note, the oocytes from extended CNP-based biphasic IVM showed the highest ROS levels (Figure 4A,B). In addition, the mRNA levels of antioxidant enzymes, i.e., *SOD*, *CAT*, *GPX4* and *NRF2*, did not change among the groups (Figure 4C), suggesting that the transcriptional regulation of an enzyme-dependent antioxidative system may not participate in the CNP-induced resistance to oxidative stress in porcine oocytes. 

Considering that increased intraoocyte ROS accumulation may induce apoptosis, we next examined the initiation of apoptosis in porcine oocytes by using annexin V-FITC staining, a well-accepted early marker of apoptosis. The results showed that the percentage of early apoptosis in the oocytes obtained from CNP-based biphasic IVM, but not extended CNP-based biphasic IVM, was significantly lower than that in oocytes undergoing conventional IVM. (Figure 4D,E). Moreover, expression analyses of apoptosis-related genes showed that the extended biphasic IVM stimulated *CASPASE-3* expression, which always occurs in the late phase of apoptosis (Figure 4F). This finding, together with the increased ROS accumulation, suggested that the extended biphasic IVM system may impair oocyte quality. 

### 3.5. CNP Reduces DNA Damage and Apoptosis in Cumulus Cells

In addition to the apoptotic status in oocytes per se, the apoptotic status of cumulus cells surrounding the oocytes has also been reported to correlate with the developmental potential of the oocytes [25]. Thus, we used a TUNEL assay to evaluate apoptotic DNA fragmentation in the cumulus cells. Porcine COCs were cultured with CNP at different concentrations for 24 h. The results showed that the DNA damage in the cumulus cells was dramatically decreased due to CNP exposure (Figure 5A,B). In addition, CNP exposure significantly downregulated pro-apoptotic genes, i.e., *BAX*, *CASEPASE3*, *C-MYC*, and *P53* (Figure 5C). Next, we analyzed apoptosis in the cumulus cells at the end of different CNP-based biphasic IVM systems. Compared with those undergoing conventional IVM, the cumulus cells from the CNP-based biphasic IVM system, but not the extended CNP-based biphasic IVM system, showed significantly lower apoptotic rates (Figure 5D,E). In line with this, the improved strategy significantly downregulated pro-apoptotic genes, i.e., *BAX*, *C-MYC*, *P53,* and the ratio of *BAX/BCL2*, a key indicator in susceptibility of the cells to apoptosis, in the expanded cumulus cells after oocyte maturation (Figure 5F,G). These results indicated that CNP may be effective in enhancing the anti-apoptotic activity and maintaining the DNA integrity of cumulus cells, thus improving oocyte maturation.

### 3.6. CNP Does Not Affect Epigenetic Modifications of In Vitro Matured Porcine Oocytes

In addition to the nuclear genome integrity, epigenetic modifications are also essential for oocyte quality and subsequent embryogenesis [27,33]. We next examined some well-established hallmark epigenetic modifications in matured oocytes undergoing different CNP-based biphasic IVM systems. The expression of oocyte-related DNA methyltransferases and demethylases mRNA were first examined by RT-qPCR analyses. Our result showed that there was no significant difference in *DNMT1*, *DNMT3A*, *DNMT3B,* and *TET3* mRNA expression levels among the groups (Figure 6A). Moreover, in line with this, the global 5mC and 5hmC levels were not changed in matured porcine oocytes due to either CNP-based biphasic IVM or extended CNP-based biphasic IVM (Figure 6B,D). 

Global transcription repression in matured oocytes was reported to be a prerequisite for normal embryogenesis after fertilization [34]. DNA methylation and repressive histone modifications (H3K9me3, H3K27me3) play essential and coordinated roles in gene silencing. Therefore, we next examined whether CNP is beneficial for the establishment of repressive histone modifications. Expression levels of relevant writers and erasers of H3K9me3 and H3K27me3 were detected by RT-qPCR analysis. The results showed that CNP did not affect either H3K9me3 or H3K27me3 writers and erasers (Figure 6F). The measurement of global H3K9me3 and H3K27me3 showed that H3K9me3 levels were not affected by CNP-based biphasic IVM or extended CNP-based biphasic IVM (Figure 6G,H), but H3K27me3 was significantly decreased in matured porcine oocytes due to extended IVM (Figure 6I,J), also supporting the possible adverse effect of extended IVM.

## 4. Discussion

The previous study by our research group has showed that pretreatment with CNP before conventional IVM can improve the developmental potential of in vitro matured bovine oocytes; thus, a novel strategy named the NFSOM system was proposed [11]. Although a pre-IVM phase relying on chemical-extended meiotic arrest has been widely used over past decades, the safety of this method is of primary concern, and evident adverse effects have been frequently reported, e.g., butyrolactone I caused degeneration of cortical granules and peripheral migration of all cytoplasmic organelles, as well as abnormal nuclear morphology [35]; roscovitine disrupted the integrity and subsequent expansion of the cumulus cells and led to mitochondrial swelling in the oocyte cytoplasm [36]; both 6-Dimethylaminopurine (6-DMAP) and cycloheximide induced notable chromosomal abnormality during subsequent embryonic development [37], etc. In contrast, CNP is a natural follicular physiological factor for maintaining meiotic arrest, thus exhibiting advantages over chemical inhibitors in regards to safety and efficacy. Supporting this, the NFSOM system dramatically increased the subsequent developmental rate of IVF bovine embryos [11]. Thus, the strategy of CNP pre-IVM has been well-accepted and widely used to update the current IVM systems in many species [11,17,18,19,20]. However, the efficiency and feasibility of CNP-based biphasic IVM system remained unsatisfactory in regards to porcine oocytes. 

In the present study, we showed that CNP-based biphasic IVM significantly improved the developmental potential of porcine oocytes, suggesting that the beneficial effect of the CNP pre-IVM strategy is common to various species. However, it should also be noted that the detailed protocol varies among species: in addition to the differences in the duration of the pre-IVM phase, the necessity of extending the IVM phase is totally distinct. In the previous study by our research group [11], using our authorized patent [38], the extended IVM phase, similar to the method used for mouse and sheep oocytes, is indispensable for improving bovine oocyte maturation [17,19]; in contrast, both our own result and previous published data [21] indicate that the extended IVM phase seemed to be detrimental to porcine oocytes; thus, the non-extended system, like that used for goat oocytes, seems to be more suitable [15] (Figure 7). Therefore, our own studies, together with those of other researches, indicate that the CNP pre-IVM strategy is definitely beneficial to oocyte quality, but the suitable technical parameters need to be further determined. Further in-depth studies are needed to explained why porcine oocytes are intolerant towards the extended IVM. A possible explanation may be related to the relatively higher intraoocyte lipid contents in porcine oocytes, which make them highly sensitive to ROS-induced impairments [39]; another possibility may be due to the much longer IVM duration of porcine IVM than those of other livestock species [22].

Having confirmed the beneficial effect of CNP-based biphasic IVM, we next attempted to understand its potential mechanisms. In addition to reduced ROS levels and inhibited early apoptosis in oocytes per se, we also focused on the cumulus cells which surround and support the oocytes. Cumulus cells play vital roles in nurturing oocyte growth, maintaining meiotic arrest, and supporting oocyte cytoplasmic maturation [23]. It has been confirmed that the degree of cumulus apoptosis was negatively correlated with the quality of matured in vitro oocytes [25]. However, whether or not CNP-based biphasic IVM can inhibit apoptosis in cumulus cells has not yet been determined. Our results indicated that CNP-based biphasic IVM enhanced the anti-apoptotic activity and maintained the DNA integrity of the cumulus cells, partly explaining the beneficial effect of CNP in improving the developmental potential of in vitro matured porcine oocytes. Interestingly, it has been reported that abnormalities in cumulus formation were found in the follicles of the *Npr2*-mutant mice [40]. In line with this, the activation of the cGMP pathway, the downstream pathway of CNP-NPR2 signaling, promoted granulosa cell survival by suppressing apoptosis in cultured preantral follicles [41]. Based on our own results and these findings, we speculate that the occurrence of apoptosis in in vitro cumulus cells may be induced by insufficient CNP-NPR2 signaling. To the best of our knowledge, this is the first report that CNP inhibits the apoptosis of ovarian cumulus cells, but further studies are needed to gain in-depth understanding of the underlying mechanisms of this result. 

The global transcriptional silencing that occurs in the oocyte genome prior to the resumption of meiosis is crucial to post-fertilization developmental transition, involving DNA methylation and histone modifications [29,42,43]. Despite the essential role of CNP in regulating the oocyte meiotic progression and nuclear maturation, the possible effect of CNP on nuclear epigenetic modifications has not been explored. However, unexpectedly, no obvious changes were detected using our model. This suggests that the beneficial role of CNP in enhancing the developmental potential of porcine oocytes may be independent in regards to the establishment of epigenetic modifications. However, the global level of H3K27me3 was significantly reduced in oocytes undergoing extended IVM, further supporting the notion that suitable technical parameters of biphasic IVM system should be carefully screened to avoid possible adverse effects. 

In summary, in the present study, we confirmed the efficiency and feasibility of the CNP pre-IVM strategy to improve the porcine oocyte IVM system (Figure 8). However, distinct from the well-established CNP-based biphasic IVM system, the extended IVM phase is not necessary, but even detrimental to porcine oocytes, emphasizing the importance of exploring the suitable technical parameters of the pre-IVM strategy among species. Concerning the mechanisms relevant to the beneficial effect of CNP pre-IVM, our results suggest that in addition to oocytes per se, the enhanced anti-apoptosis and resistance to oxidative stress in the cumulus cells may contribute considerably to improved oocyte quality, which has been largely overlooked in previous studies.

## Figures and Tables

**Figure 1 genes-14-01885-f001:**
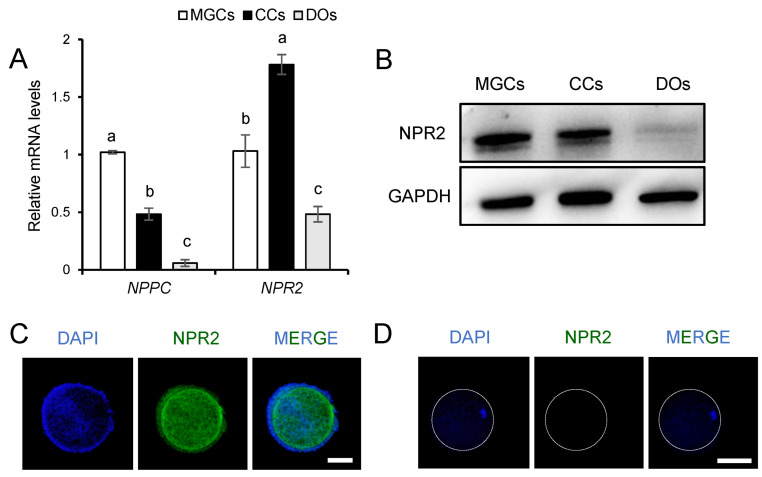
Quantification of CNP and NPR2 expression in porcine MGCs, CCs, and DOs. (**A**) Detection of *NPPC* and *NPR2* by RT-qPCR in isolated MGCs, CCs, and DOs. (**B**) Western blot analysis revealed the presence of NPR2 protein in porcine MGCs and CCs, but not in DOs. (**C**) NPR2 protein in porcine COCs confirmed by fluorescence analysis; scale bar, 100 μm. (**D**) Detection of NPR2 protein in porcine DOs by fluorescence analysis; scale bar, 100 μm. ^a–c^ Values indicated by different letters are significantly different, *p* < 0.05.

**Figure 2 genes-14-01885-f002:**
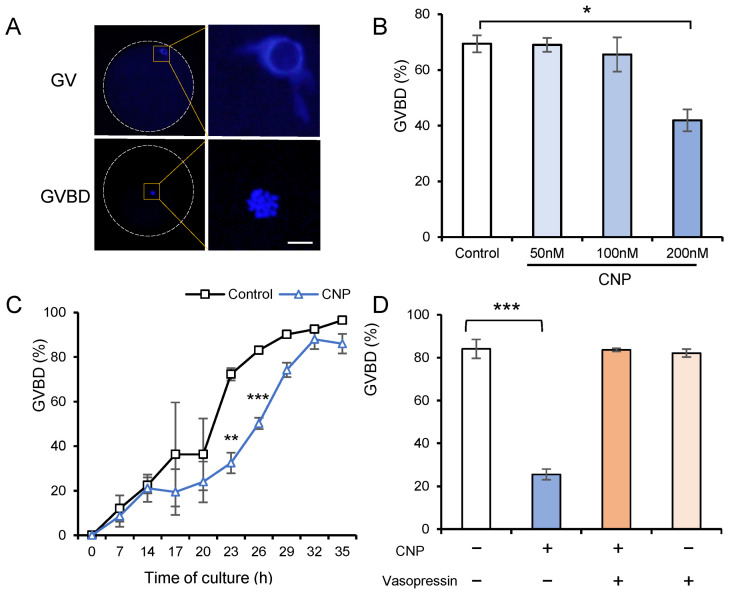
The effect of CNP on the progression of nuclear maturation in porcine oocytes in vitro. (**A**) Representative images of chromatin configurations of porcine oocytes (blue) in different maturation stages (GV: all chromatin is assembled into a compact nuclear circle confined around the nucleolus; GVBD: exhibits a small nucleus and arranged condensed chromosomes). Chromatin was stained with DAPI. Scale bar, 10 μm. (**B**) GVBD rate in porcine CC-enclosed oocytes cultured in medium supplemented with 50, 100, 200 nM CNP for 24 h. (**C**) Kinetics of meiotic resumption of CC-enclosed oocytes cultured in medium supplemented with 200 nM CNP. (**D**) Effect of NPR2 inhibitor (vasopressin) on the nuclear maturation of COCs cultured in medium supplemented with 200 nM CNP for 24 h. * *p* < 0.05; ** *p* < 0.01; *** *p* < 0.001.

**Figure 3 genes-14-01885-f003:**
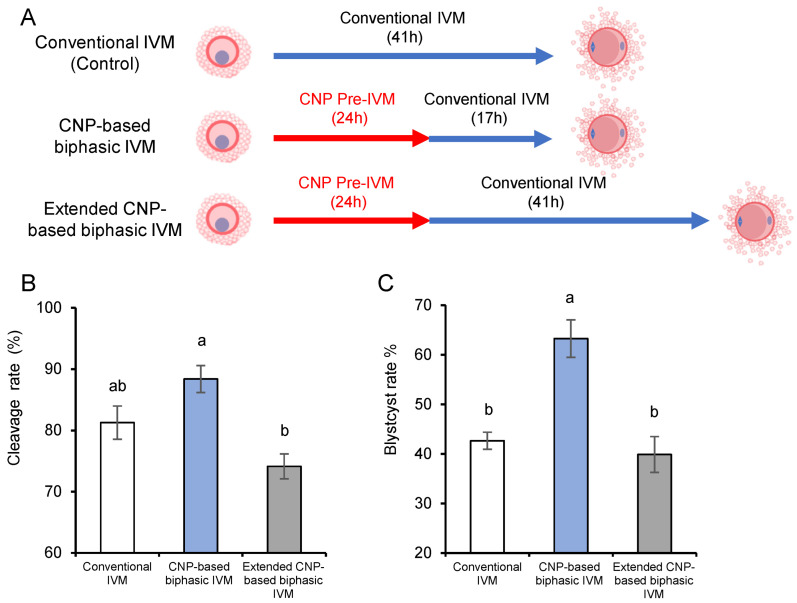
The effect of the CNP-based biphasic IVM system on the development rate of porcine oocytes after parthenogenetic activation. (**A**) Model illustrating the methodology of the CNP-based biphasic IVM system, which consists two steps: a pre-IVM phase based on CNP supplementation, followed by a 17 h or 41 h conventional IVM phase. (**B**,**C**) The effect of CNP-based biphasic IVM on the subsequent cleavage rate (**B**) and blastocyst rate (**C**) after parthenogenetic activation. ^a,b^ different letters indicate significant differences; *p* < 0.05.

**Figure 4 genes-14-01885-f004:**
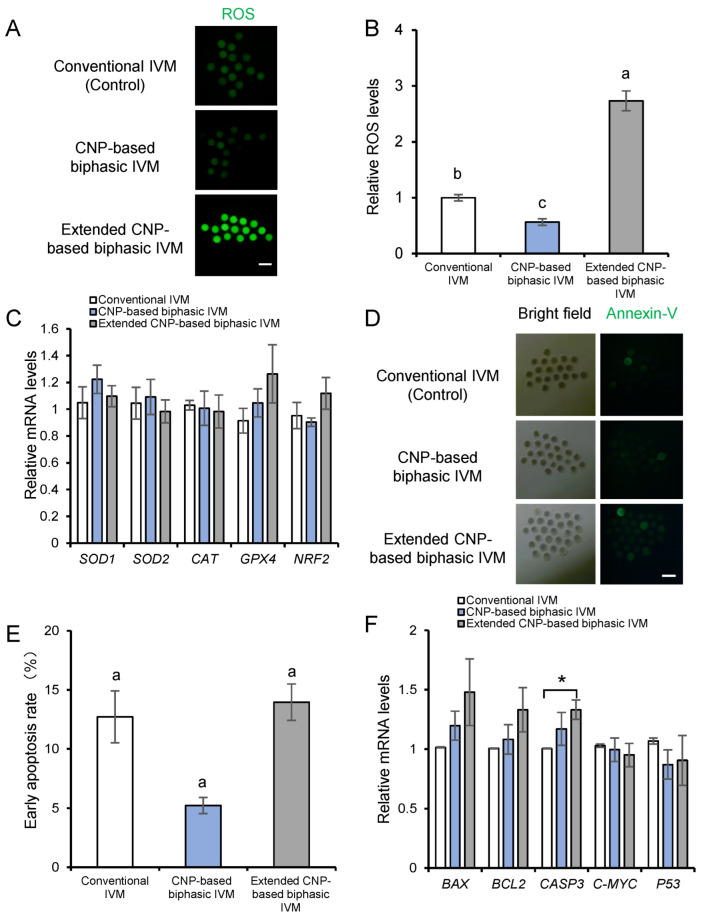
CNP-based biphasic IVM alleviated ROS accumulation and apoptosis initiation in porcine oocytes. (**A**) Illustrative images of DCHF-DA fluorescence (green) in MII oocytes from each group; scale bar, 200 μm. (**B**) Quantitative analysis of ROS fluorescence intensity in MII oocytes from the control IVM (*n* = 40), CNP-based biphasic IVM (*n* = 38), and extended CNP-based biphasic IVM (*n* = 40) groups. (**C**) Relative mRNA expressions of antioxidant enzyme genes in the oocytes among the groups. (**D**) Representative images of annexin-V fluorescence (green) in MII oocytes from each group; scale bar, 200 μm. (**E**) The percentage of annexin-V positive oocytes from each group. (**F**) Relative mRNA expressions of apoptosis-related genes in the oocytes. Experiments were conducted in triplicate, with more than 50 oocytes examined for each group. ^a–c^ Values indicated by different letters are significantly different; *p* < 0.05. * *p* < 0.05.

**Figure 5 genes-14-01885-f005:**
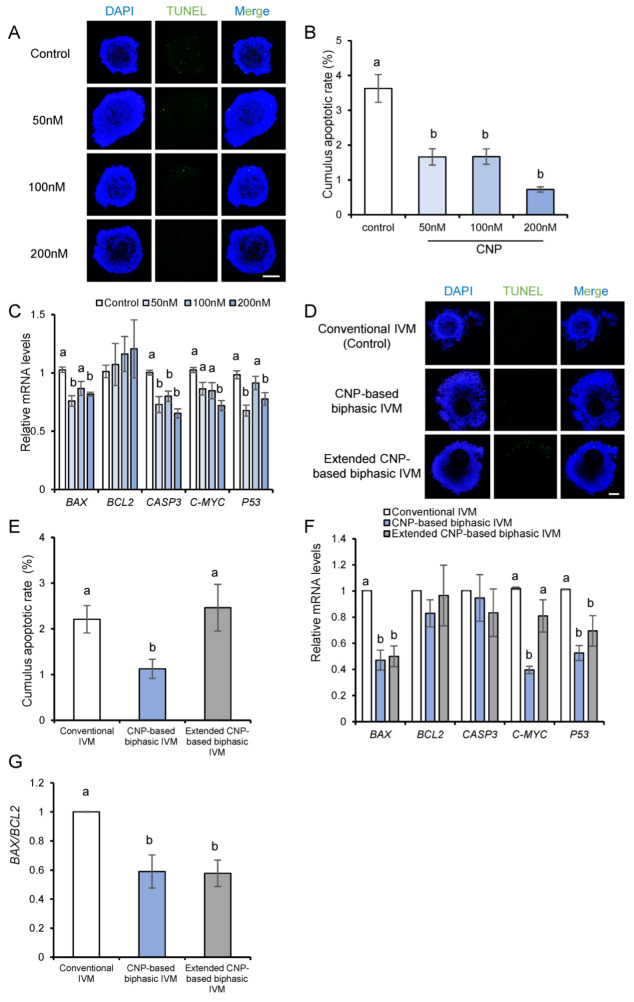
CNP reduced DNA damage and apoptosis in cumulus cells. (**A**) Representative images of TUNEL (green) in COCs from different concentrations of CNP (0, 50, 100, 200 nM); scale bar, 200 μm. (**B**) The percentage of TUNEL positive cumulus cells from the control (*n* = 61), 50 nM CNP (*n* = 40), 100 nM CNP (*n* = 44), and 200 nM CNP (*n* = 30) groups. (**C**) Relative mRNA expression of apoptosis-related genes, i.e., *BAX*, *BCL2*, *CASPASE-3*, *C-MYC*, *P53*, in cumulus cells. (**D**) Representative images of TUNEL (green) in COCs from each group; scale bar, 100 μm. (**E**) The percentage of TUNEL positive cumulus cells from the control IVM (*n* = 62), CNP-based biphasic IVM (*n* = 56), and extended CNP-based biphasic IVM (*n* = 38) systems. (**F**) Relative mRNA expression of apoptosis-related genes, i.e., *BAX*, *BCL2*, *CASPASE-3*, *C-MYC*, and *P53*, in cumulus cells after IVM. (**G**) The ratio of *BAX/BCL2* in each group; ^a,b^ Values with different letters are significantly different; *p* < 0.05.

**Figure 6 genes-14-01885-f006:**
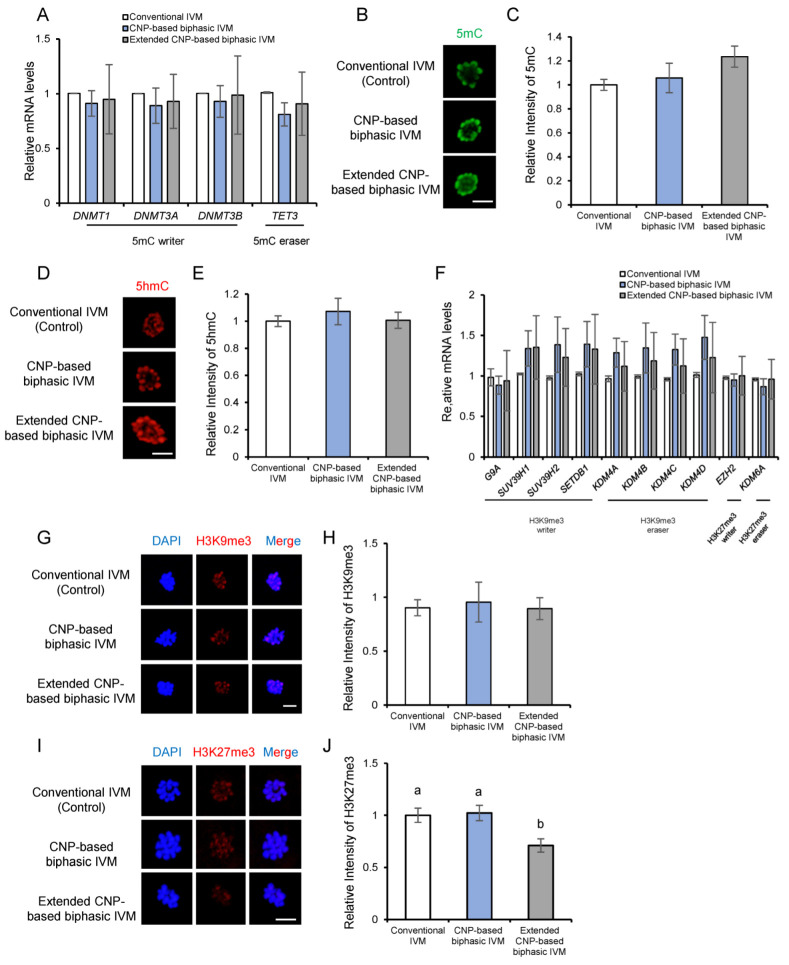
CNP does not affect the epigenetic modifications of in vitro matured porcine oocytes. (**A**) Representative images of 5mC (green) in MII oocytes from each group; scale bar, 10 μm. (**B**) Quantification of relative levels of 5mC in mature oocytes from the control (*n* = 67), CNP-based biphasic IVM (*n* = 38), and extended CNP-based biphasic IVM (*n* = 60) groups; (**C**) Representative images of 5hmC (red) in MII oocytes from each group; scale bar, 10 μm. (**D**) Quantification of the relative levels of 5hmC in mature oocytes from the control (*n* = 59), CNP-based biphasic IVM (*n* = 41), and extended CNP-based biphasic IVM (*n* = 68) groups. (**E**) Expression of DNMTs and TET3 mRNA levels in oocytes. (**F**) Representative images of H3K9me3 marks (red) and nuclear DNA (blue) in MII oocytes from each group; scale bar, 10 μm. (**G**) Quantification of relative levels of H3K9me3 in mature oocytes from the control (*n* = 57), CNP-based biphasic IVM (*n* = 43), and extended CNP-based biphasic IVM (*n* = 59) groups; (**H**) Typical images of H3K27me3 (red) and nuclear DNA (blue) in MII oocytes from each group; scale bar, 10 μm. (**I**) Quantification of relative levels of H3K27me3 in MII oocytes from control IVM (*n* = 60), CNP-based biphasic IVM (*n* = 44), and extended CNP-based biphasic IVM (*n* = 55) groups. (**J**) Expression of writer and eraser mRNA levels of H3K9me3 and H3K27me3 in oocytes. Experiments were repeated at least 3 times, with more than 50 oocytes examined for each experimental condition. ^a,b^ Values indicated by different letters are significantly different; *p* < 0.05.

**Figure 7 genes-14-01885-f007:**
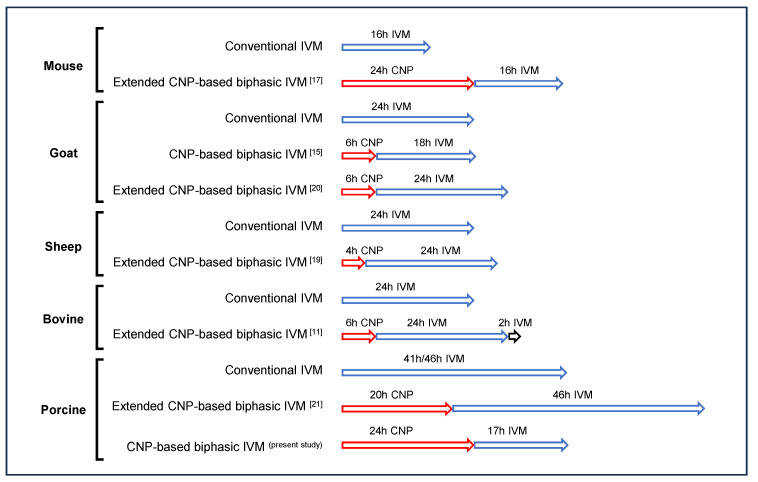
Model illustrating the methodology of the CNP-based biphasic IVM system of different species (mouse [17], goat [15,20], sheep [19], bovine [11], porcine [21]), which consist of two steps: a pre-IVM phase based on CNP treatment, followed by an extended or un-extended IVM phase.

**Figure 8 genes-14-01885-f008:**
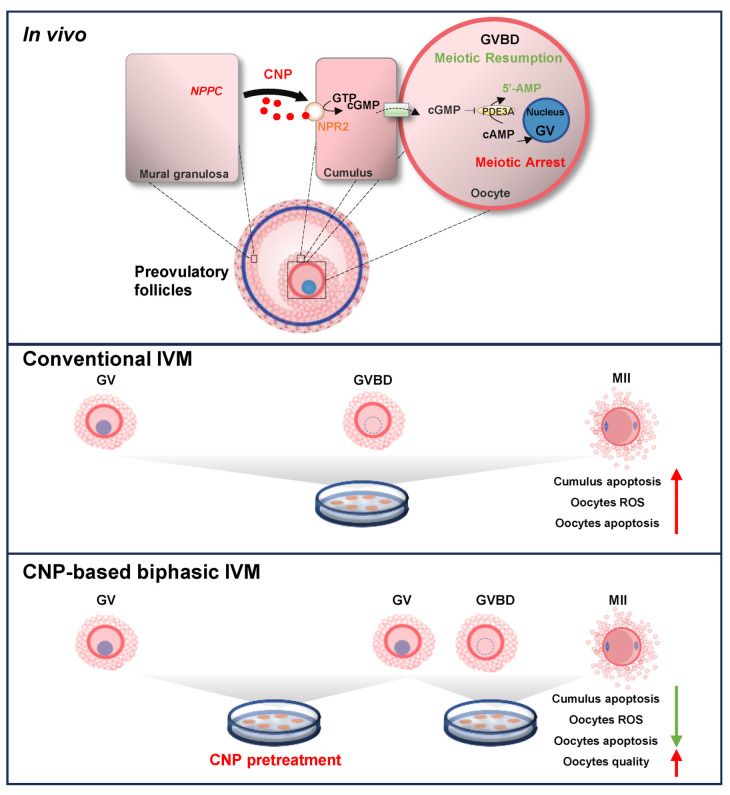
A model illustrating the importance of the CNP-based biphasic IVM system. In the growing follicles, CNP secreted from the mural granulosa cells acts on its specific receptor NPR2 located on the cumulus cells, thus resulting in the maintenance of meiotic arrest until LH-induced oocyte meiotic resumption. In vitro isolation of the oocytes triggers spontaneous and precocious meiotic resumption, ultimately compromising the developmental potential of the oocytes. In the CNP-based biphasic IVM system, exogenous CNP supplementation into the traditional medium not only inhibits the spontaneous meiotic resumption of COCs, but also reduces the apoptosis of cumulus, decreases the ROS and apoptosis of the oocytes, and improves the quality of the MII oocytes.

**Table 1 genes-14-01885-t001:** List of primer sequences.

Gene Name	Forward Primers	Reverse Primers
*GAPDH*	TCGGAGTGAACGGATTTG	CCTGGAAGATGGTGATGG
*RPL19*	GGAAGGGTACTGCCAATGCT	GTGCTCCATGAGAATCCGCT
*CNP*	CCGAAGGTCCCTCGAACTC	GGAGTCTTGTCGCCCTTCTT
*NPR2*	AATGGAGTCTAACGGTCAAG	GGAAGAAGTAGGGTTTATAGGA
*DNMT1*	GCGTCTTGCAGGCTGGTCAGTA	CTTCTTATCATCGACCACGACGCT
*DNMT3A*	ATCAGTACGACGATGACGGC	CACCAAGAGATCCACGCACT
*DNMT3B*	ACCTGTCATCCGACACCTCT	CTCGGCATGAACCCACGTTA
*TET3*	TCTTCCGTCGTTCAGCTACTACAG	GTGGAGGTCTGGCTTCTTCTCAAA
*G9A*	GGAGGAGCTGGGGTTTGAC	CAGAGGTGGCTGCTGAGTTG
*SUV39H1*	GAATCAGCTCCAGGACCTGTGC	CAGGTGCTCTCTGAGTCTGGGTAC
*SUV39H2*	GCAGGACGAACTCAACAGAA	CAACCAAAGGTGGCTTCATT
*SETDB1*	CATTGGTTTGGATGCAGCAGC	GATGCATCATCAAAGAGCTGGTC
*KDM4A*	CTGAAACCTTGAACCCCAGTGC	GATATCGTCATAGGATGCCCGTG
*KDM4B*	CTGGCCAACAGCGAGAAGTACTG	GATGTTCCACTGGGCCACGTC
*KDM4C*	TGTGAAAAGCCAGGAGAAGCAAAG	CAGGTTTGGTCAGCCTCGGT
*KDM4D*	AAGGATGCAGTGTGTGTTGC	CCTGTTCGCGGATCTTTTTA
*EZH2*	TGCAACACCCAATACTTACAAGC	ACTCTTTTGCTCCCTCCAAGT
*KDM6A*	GCAGGCTCAGTTGTGTAACC	GGTTTACATGCCTGCTGTGC
*BAX*	TGCCTCAGGATGCATCTACC	AAGTAGAAAAGCGCGACCAC
*BCL2*	AATGACCACCTAGAGCCTTG	GGTCATTTCCGACTGAAGAG
*CASPASE-3*	CCGAGGCACAGAATTGGACT	TCGCCAGGAATAGTAACCAGG
*C-MYC*	GATAGTGGAAAACCCGGCTGC	CAGATATCCTCGCTGGGTGC
*P53*	TTTCACCCTCCAGATCCGTG	TTTATGGCGGGAGGGAGACT
*SOD1*	TCCATGTCCATCAGTTTGGA	AGTCACATTGCCCAGGTCTC
*SOD2*	AAGCCATCAAACGCGACTTT	CCTTGTTGAAACCGAGCCAA
*CAT*	ACATGGTCTGGGACTTCTGG	TCATGTGCCTGTGTCCATCT
*GPX4*	ATTCTCAGCCAAGGACATCG	CCTCATTGAGAGGCCACATT
*NRF2*	CATAGCAGAGCCCAGTACCA	CACGGTGGTCTTGGTTGAAG

## Data Availability

The data that support the findings of this study are available on request from the corresponding author.

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
