# Peer review of "A Redesigned Method for CNP-Synchronized In Vitro Maturation Inhibits Oxidative Stress and Apoptosis in Cumulus-Oocyte Complexes and Improves the Developmental Potential of Porcine Oocytes"

_genes, 2023, doi:10.3390/genes14101885_

Round 1

Reviewer 1 Report

The authors attempted to enhance the current IVM system for porcine oocytes by assessing the concentration and treatment time of CNP required to inhibit spontaneous meiotic resumption. Additionally, they observed increased expression of anti-apoptotic and anti-oxidative genes.

Major Comments

While this manuscript does provide basic data, such as the expression of NPPC and the effect of CNP on the progression of nuclear maturation, as shown in Figures 1 and 2, it appears that these findings have already been published (Theriogenology, 2018; 106:198-209, #11). Therefore, it is advisable to revise this manuscript to focus solely on presenting new discoveries.

Minor comments

1.    P2 Line45-, “Our previous study” refers to reference #11, which seems not to be the authors’ work.  

Reviewer 2 Report

- Writing Mistakes:

1) line 40: the point instead of the comma

2)line 298: missing the point before the word "Moreover"

- Figure Mistakes:

1) Figure 2 (A): describe in more details the figure, i.e. the coloration 

Reviewer 3 Report

This paper aims to evaluate the role C-type natriuretic peptide (CNP) in maintaining meiotic arrest in IVF, as a way to improve in vitro maturation. In addition to evaluating to protocols to use CNP in porcine in vitro maturation, I also explores the role of the cumulus cells in the mechanisms by which the CNP have beneficial effects in the in vitro maturation rates.  The team has already approach this peptide in in vitro maturation in the bovine, in a published study, and is now further exploring its use in the porcine. The paper is generally well written and well organized, and I have only a few specific correction.

Line 23 : “may be largely divergent among species and need to be redesigned specifically” – change to needs

Line 45 – “. Our previous study has established (…) – state the species

Line 66 – “led us to ask the detailed method may” - …led us to question if the detailed method may…

In the immunofluorescence staining, please provide the references for secondary antibodies

Line 177 - Total RNA was extracted from MGCs, CCs, DOs using TRizol (Invitrogen) according to the manufacturer’s instructions – did you use pools or individual COCs?

In figure 7 – in the line about the bovine, it says only VM
